# Multi-Object Detection in Security Screening Scene Based on Convolutional Neural Network

**DOI:** 10.3390/s22207836

**Published:** 2022-10-15

**Authors:** Fan Sun, Xiangfeng Zhang, Yunzhong Liu, Hong Jiang

**Affiliations:** College of Intelligent Manufacturing and Industrial Modernization, Xinjiang University, Urumchi 830017, China

**Keywords:** security screening scenes, convolutional neural networks, multi-scale feature extraction, attentional mechanisms

## Abstract

The technique for target detection based on a convolutional neural network has been widely implemented in the industry. However, the detection accuracy of X-ray images in security screening scenarios still requires improvement. This paper proposes a coupled multi-scale feature extraction and multi-scale attention architecture. We integrate this architecture into the Single Shot MultiBox Detector (SSD) algorithm and find that it can significantly improve the effectiveness of target detection. Firstly, ResNet is used as the backbone network to replace the original VGG network to improve the feature extraction capability of the convolutional neural network for images. Secondly, a multi-scale feature extraction (MSE) structure is designed to enrich the information contained in the multi-stage prediction feature layer. Finally, the multi-scale attention architecture (MSA) is fused onto the prediction feature layer to eliminate the redundant features’ interference and extract effective contextual information. In addition, a combination of Adaptive-NMS and Soft-NMS is used to output the final prediction anchor boxes when performing non-maximum suppression. The results of the experiments show that the improved method improves the mean average precision (mAP) value by 7.4% compared to the original approach. New modules make detection much more accurate while keeping the detection speed the same.

## 1. Introduction

X-ray security screening is now widely used in the subway, airports, train stations, campuses, superstores, and other scenarios. At the same time, with the continuous development of artificial intelligence technology, intelligent security solutions are gradually gaining widespread attention in the industrial sector. However, at the moment, contraband detection under the X-ray security machine relies primarily on manual discrimination. While the X-ray image background is complicated, many targets, low color contrast, and occlusion phenomenon are more serious [1], and the security personnel needs to spend a long time on the detection process.

Alex et al., introduced the AlexNet [2] model, which employs deep convolutional neural networks to solve image classification problems and is significantly more effective than conventional image processing algorithms, demonstrating the superior performance of deep neural networks in the field of computer vision. Since then, the application of deep convolutional neural networks to image recognition, image segmentation, and target detection problems has become an essential topic of study. Currently, target detectors are categorized into one-stage target detectors and multi-stage target detectors based on the configuration of their networks.

Girshick [3] et al., initially presented the R-CNN network model to extract features using convolutional neural networks as an alternative to classic target detection approaches using artificially created feature extraction and description approaches. After that, convolutional neural network technology experienced rapid growth. With the development of artificial intelligence, a significant number of outstanding target detection algorithms have appeared, all of which employ convolutional neural networks to handle target detection challenges. Girshick [4] et al., introduced the Fast-RCNN network architecture to perform convolution on the whole image, ROI Polling to generate fixed-size feature maps, and Softmax instead of SVM classifier to increase target detection network speed and accuracy. Ren [5] et al., proposed Faster-RCNN to produce region recommendations via RPN. The RPN framework improves the network’s speed of detection. Even though the multi-stage detector can achieve high detection accuracy, its detection speed is exceedingly slow.

Redmon [6] et al., proposed the YOLO object detection framework, a neural network framework appropriate for targets detected in real-time and with a detection speed of 45 FPS for the first time. Later versions of YOLO, YOLO 9000, YOLO v3, YOLOv4, and YOLOv7 [7,8,9,10] included new neural network architectures such as batch normalization [11], FPN [12], and SPP [13] to strike a balance between detection accuracy and speed. To build a genuinely anchorless detector, Tian [14] et al., suggested Fully Convolutional One-Stage Object Detection (FCOS) based on the RetinaNet [15] network architecture. The Single Shot MultiBox Detector (SSD) was proposed by Liu [16] et al., as an anchor-frame-based single-stage target detector that obtains the first *mAP* value of a single-stage target detector above a multi-stage detector. One-stage detection may reach a relatively high speed and can frequently achieve acceptable accuracy on a large scale. The problem is that it is difficult to achieve good detection accuracy for small objects, particularly when there are targets of varying scales in the same image.

Through the literature [17], we can compare the detection speeds of current mainstream target detectors. Single-stage target detectors can infer 24–105 times faster than multi-stage target detectors. The fastest single-stage target detectors detect speeds can up to 200 FPS, while the fastest multi-stage target detectors only reach 5 FPS. Because the object detection algorithm in this paper needs to be applicable to HIKVISION's ISD-SC5030SA-1C type X-ray security screening equipment, the following conditions need to be considered. The screening machine is suitable for daily scenarios of baggage and parcel security inspection and uses a line-by-line scanning method for imaging. In order to ensure that the imaging results are clear and stable, the maximum transmission speed of the screening machine is specified as 0.2 m/s. In order to ensure the efficiency of the inspection, it is necessary to have a passing rate of at least 720 items/h. The maximum size of the screening machine allowed through is 500 mm × 300 mm (width and height). The maximum length of the package allowed to be detected is 980 mm. Taking into account the above conditions, X-ray security screening requires at least 10 FPS detection speed to meet the detection requirements, making it challenging for multi-stage target detectors to meet real-time detection requirements. The original SSD algorithm used in this paper can reach a minimum detection speed of 19 FPS, which meets all the requirements, and the improved SSD algorithm can reach an average detection speed of 22 FPS. 

For X-ray image target detection, Qiao [18] et al., employed pyramidal convolution to improve multi-scale feature extraction for X-ray image target recognition and introduced band pooling to use image contextual information to improve detection accuracy. Zhang [19] et al., used the DenesNet [20] network for feature fusion to improve network feature extraction and characterization. Combining multi-scale contextual information, Wu [21] et al., applied dilation convolution to enhance detection accuracy. Akacy [22] et al., employed regional full convolutional neural networks to identify X-ray image targets. Galvez [23] et al., used a transfer learning approach to address the X-ray image classification problem, which significantly increased classification accuracy. The above works focus on improving neural networks’ ability to extract and characterize image features. However, they do not consider that a lot of redundant background information is extracted when extracting features. Redundant background information can make it harder for target classification and boundary regression networks to find objects.

The literature [24] uses transfer learning to detect small-size targets in X-ray images using the Faster-RCNN-based fusion feature pyramid network (FPN) for multi-scale feature adaption and localization of small-size targets utilizing a small anchor point strategy and ROI pooling method. Literature [25] suggested a cascaded structure tensor technique for acquiring contour-based suggestions, removing clutter and occlusion from X-ray images, and then passing the candidate proposals to a single feed-forward convolutional neural network for recognition. In the literature [26], in order to address the occlusion problem in X-ray image detection, a de-obscuring attention module (DOAM) and diverse materials exhibiting visually distinct colors and textures are utilized to build an attention tensor and enhance the detector’s feature map information. Literature [27] suggests a lateral inhibition module (LIM) that maximally inhibits the flow of noisy information via a bidirectional propagation (BP) module and activates the most attractive borders from four directions via a boundary activation (BA) module. Literature [28] employs adversarial domain adaption approaches to match the background distribution of a large number of unlabeled SoC samples. This approach helps to train the network to detect items in the SoC dataset using a small labeled dataset. The previous work focused on solving the problems of dense stacking and mutual occlusion of X-ray images or making a better dataset of X-ray images. However, they do not consider the limitations of existing fixed-size convolution kernels to handle widely available objects with different scales.

In recent years, NVIDIA has continued to launch higher-performance GPU products, and the problem of arithmetic power limiting the development of deep learning has also been alleviated to some extent. In industrial scenarios, SSD has been able to meet real-time requirements, while compared to the YOLO series of continuous updates, up to YOLOv7, SSD algorithm updates and optimization have been relatively limited. However, considering background feature redundancy has a substantial effect on model detection performance, and the feature pyramid structure has been demonstrated to play a crucial role in multi-scale and small object detection in target detection tasks. The SSD architecture prediction feature layer essentially acts as a divide-and-conquer for different scales [29]. Hence, the focus of this article is on SSD-based architectures, which comprise:(1)Replacing the VGG [30] architecture in the original literature with the ResNet [31] architecture to enhance the backbone network model extraction and characterization capabilities.(2)Based on the divide-and-conquer concept, we create a multi-scale extraction module to enrich the information inside the multi-stage prediction feature layer.(3)On the feature prediction layer, a multi-scale attention mechanism is combined to get rid of redundant feature interference and pull out useful context information.

The other parts of this paper are arranged as follows: Section 2 introduces the basic idea and network structure of the classical algorithm SSD; Section 3 illustrates the improvement strategy of improved SSD in detail; Section 4 presents the experimental environment, dataset distribution, experimental result analysis, ablation experiment, and algorithm comparison. Finally, Section 5 provides the conclusions, and Section 6 introduces follow-up work and improvement direction.

## 2. Materials and Methods

### 2.1. Original Network Architecture

SSD is the first one-stage anchor-based target detection method that outperforms multi-stage target detection in terms of precision and inference time. Firstly, using the VGG network to extract image features and additional layers, the multi-scale feature prediction layers are added to predict results, and non-maximum suppression is used to generate the final detectable results. The core idea is mainly to predefine multi-scale anchor boxes on the prediction feature layer, later to obtain the detected target’s location by regressing the anchor boxes, and at the same time, to classify the class of the corresponding target. The specific steps include the following.

#### 2.1.1. Backbone Network

Numerous experiments have shown that the depth of a convolutional neural network significantly impacts the performance of tasks such as image classification, target detection, image segmentation, and so on. As the network’s depth rises, the model can extract richer semantic information and perform better in practical tasks. The VGG network investigates the effect of neural network depth on model performance for the first time using a 3 × 3 small receptive field convolutional filter. While MaxPooling layers 5 provide spatial pooling with step size 2 and window size 2 × 2, the linear transformation of the input channels is achieved by 1 × 1 convolution layer, replacing the fully connected layers. Figure 1 depicts the architecture of the VGG network. The VGG network outperforms the neural network with higher breadth but fewer layers for the same number of parameters. As a result, the original SSD method used the VGG architecture as the backbone network.

The SSD approach employs the VGG-16 network architecture as the backbone network. Still, it retains only the first 15 convolution layers of its feature extraction part while adjusting the final MaxPooling5 downsampling from the original network with a step size of 2 and a window size of 2 × 2 to a step size of 1 and a window size of 3 × 3. Following the adjustment, the size of Conv4 and Conv5 feature maps will remain unchanged. Finally, the VGG-16 model’s fully connected layers FC6 and FC7 are replaced by 3 × 3 convolution layers (Conv6) and 1 × 1 convolution layers (Conv7). The Conv4_3 convolution layer serves as the first feature prediction layer, and Conv6_2 serves as the second feature prediction layer. Figure 2 depicts the architecture of the SSD backbone network.

Four extra feature prediction layers are introduced as additional layers in the SSD approach to adapt to detecting targets of varied scales. The subsequent layer employs one 1 × 1 convolution layer to reduce the channel dimension, then adds one 3 × 3 convolution layer to refine semantic contexts. All convolutional operations are followed by batch normalization and ReLU activation functions. The additional feature prediction layers have 256-dimensional output channels, except for Conv8_2, which has 512-dimensional output channels. The feature map sizes of the various feature prediction layers are 10 × 10, 5 × 5, 3 × 3, 1 × 1, respectively. Four 3 × 3 convolution layers make up the classification and regression networks of the feature prediction layers Conv7 to Conv9, and four 1 × 1 convolution layers make up Conv10. The classification and regression networks then receive data from the feature extraction layer. For all prediction categories, the number of output channels of the classification network equals the number of predefined anchors at feature map cells on the feature map. The number of output channels for all regression networks is simply the number of four offsets of predetermined anchor frames at each feature map cell.

#### 2.1.2. Predefined Anchors and Positive and Negative Sample Matching

When predefining anchor boxes for multiple feature mapping at the network’s top, a set of bounding boxes is associated with each feature mapping cell. As the feature map’s pixel points are tiled, the position of each default anchor box relative to its related cell is fixed. Using default box shapes with varying aspect ratios and sizes in the same feature mapping layer can also efficiently match the spatial distribution of possible output box shapes. The default anchor box sizes in different feature layers vary to allow the detection of targets of varying scales. SSD sets the hyperparameters of aspect ratio to *a_r_ =* {1, 2, 3, 1/2, 1/3} respectively when predefining anchor boxes, and finally uses Equation (1) to design the sizes of different anchor boxes as follows.
(1)$$S_{k}=\frac{S_{\max}-S_{\min}}{m-1}(k-1)+S_{\min},k\in [1,m]$$

In the formula, assume that the prediction feature layer used to generate the default box is m and that 0.2 and 0.9 represent the scales of the default anchor box in the lowest feature prediction layer and the highest feature prediction layer, respectively. In contrast, take the arithmetic progression approach and expand the default anchor box size for the other predicted feature layers. The following expression sets the width of the default anchor box for the current feature map: $w_{a}{}^{k}=S_{r}\sqrt{a_{r}}$. The height calculation formula is as follows: $w_{a}{}^{k}=S_{r}/\sqrt{a_{r}}$, and an additional set of anchor boxes with an aspect ratio of 1 with a scale of $S_{k}{}^{\prime }=\sqrt{S_{k}S_{K}{{}_{+}}_{1}}$, and the center point of the anchor box is $\left(\frac{i+0.5}{\left|f_{k}\right|},\frac{j+0.5}{\left|f_{k}\right|}\right)$ where $\left|f_{k}\right|$ is the size of the *k-th* feature map layer. The definition of the default anchor box is still an open problem, and other clustering algorithms such as *K*-means also generate predefined anchor boxes for different tasks. The final distribution of predefined anchor frames and aspect ratios used in the SSD algorithm is shown in Table 1.

Literature [32] suggests that positive and negative sample matching, one of the key factors causing the performance gap between various target detection network models, and the severe imbalance between positive and negative samples, can significantly hinder the model’s final performance. The SSD model matches positive samples based on the existing anchor boxes and mostly follows the two rules below.
(1)Calculate the intersection over union between the ground truth boxes and the default anchor boxes. For each ground truth box, choose the intersection over union with the highest value as the positive sample.(2)Mark the remaining default anchors not marked as positive samples if their intersection over union is greater than 0.5 with any ground truth box.

Based on the aforementioned matching principle, numerous predetermined anchor boxes can be allocated to each ground truth box, thereby facilitating the selection of the anchor box with the greatest score for regression in the prediction stage. Algorithm 1 shows the algorithm flow detail. The remaining anchor boxes are designated as negative samples. Typically, the number of negative examples is significantly more than the number of positive samples. The difference between the number of positive and negative samples is a major factor that hurts network performance during learning. It can cause a significant drop in how well the network can detect targets. Therefore, the SSD algorithm only selects the default anchor box that incorrectly predicts negative samples as positive samples with the highest scores in a ratio of 1:3 of positive to negative samples to calculate the loss in the classification network. All other default anchor frames are discarded. Other default anchors are not taken into account, and negative samples are not used to figure out regression loss.
**Algorithm****1** Positive and negative sample matchingInput:    g: Real sample box, mainly including the coordinate information of the upper left and lower right corners    l: True sample label    d: default anchor box, mainly including the upper left and lower right coordinates    I_w_: the width of the image, the image width and height are equal in sizeOutput:    O_l_: the positive sample label matched by the default box    O_d_: the default anchor box as the positive sample
1. Real sample frame normalization: g = g/I_w_2. Default anchor box normalization.    For each feature map do:       d = d/I_w_3. Calculate the intersection over union of any two anchor frames:    iou = IOU (g, d)4. Get the default box matched by the real sample box and its index (first dimension):    best1_iou, index1 = max(iou(1))5. get the real box matched by the default anchor box and its index (second dimension):    best2_iou, index2 = max(iou(2))6. set index1 index to the default box intersection over union to 1.0:    best2_iou[index1] = 1.07. Modify index2 index to the real sample box index.    idx = arange(0, index2.size)    index2[index1[idx]] = idx 8. get the mask of the remaining default anchor boxes larger than the threshold:    mask = best2_iou>0.59. index of all default box positive samples:    index2 = index2[mask]10. set the default anchor box label.    O_l_[8732] = 0    O_l_[index2] = l[index2]11. Positive sample anchors:    O_d_ = d[index2]Return O_l_, O_d_

#### 2.1.3. Loss Function

The overall loss function is the average of the localization loss and the confidence loss. The default value of *a* in Equation (2) is 1. $x_{ij}{}^{p}=\{1,0\}$ is an indicator function, indicating that the *i*-th default box matches the *j*-th real sample box, and *p* indicates that the default box belongs to a category. A default box can only correspond to a ground truth box, whereas a ground truth box may contain numerous default anchor boxes. The localization loss is expressed using the *smooth_l1_* function. The advantage of the *smooth_l1_* function is that when the input is small, the loss gradient decreases, and the network converges to the correct parameters. In contrast, when the input is large, the absolute value of the loss function reaches the upper limit of 1, which is not so large as to destroy the network parameters and cause unstable training. Simultaneously, the centroid position parameters and the width and height of the actual sample frame are normalized to enable neural network learning. Equations (2)–(4) show, in order, the general loss function of the SSD method, the localization loss function, and the confidence loss function.
(2)$$L(x,c,l,g)=\frac{1}{N}(L_{conf}(x,c)+\alpha L_{loc}(x,l,g))$$
(3)$$\begin{array}{c} L_{l}{}_{oc}(x,l,g)=\sum_{i\in pos}^{N}\sum_{m\in \{cx,cy,w,h\}}x_{i}{{}_{j}}^{k}smoooth_{L1}(l_{i}{}^{m}-{\hat{g}}_{j}^{m})\\ {\hat{g}}_{j}{}^{cx}=\frac{g_{j}{}^{cx}\;-\;d_{i}{}^{cx}}{d^{w}{}_{i}}\\ {\hat{g}}_{j}{}^{cy}=\frac{g_{j}{}^{cy}\;-\;d_{i}{}^{cy}}{d_{i}{}^{h}}\\ {\hat{g}}_{j}{}^{w}=\log(\frac{g_{j}{}^{w}}{d_{i}{}^{w}})\\ {\hat{g}}_{j}{}^{h}=\log(\frac{g_{j}{}^{h}}{d_{i}{}^{h}}) \end{array}$$
(4)$$L_{conf}(x,c)=-\sum_{i\in Pos}^{N}x_{ij}{}^{p}\log({\hat{c}}_{i}{}^{p})-\sum_{i\in Neg}\log({\hat{c}}_{i}{}^{0})where\;{\hat{c}}_{i}{}^{p}=\frac{\exp(c_{i}{}^{p})}{\sum_{p}\exp(c_{i}{}^{p})}$$

## 3. Improved Approach

### 3.1. Improved Backbone

The literature [33] indicates that the overall depth of the model is very important for performance. Shallow neural networks focus more on the image’s edge contour and color information. As the network layers deepen, the neural network will focus more on texture information. As the network continues to deepen, the layers will focus on information with class specificity, object pose, and general object class information. The shallower neural network layers often contain more precise location information, whereas the deeper neural network levels contain semantic information regarding the target. Therefore, adding deeper backbone networks can greatly improve target detection accuracy. Compared to VGG-16 as the backbone network, ResNet-50 achieves a deeper convolutional neural network by reducing the number of parameters by 82.97 percent while avoiding the problem of performance degradation in the deep model via the residual connection module. Figure 3 depicts the basic module of ResNet. The 3 × 3 convolution step of Conv4_1 is changed to 1, and the 1 × 1 convolution step of residual connection is changed to 1. Consequently, the size of the Conv4 feature map is the same as that of Conv3, and five more feature prediction modules comprised of 1 × 1 convolution layer and 3 × 3 convolution layer are added. The output of all convolution layers is batch normalized and nonlinearly activated by the ReLU function while the step size of all 1 × 1 convolution layers is set to 1, the step size of all 3 × 3 convolution layers is {2,2,2,1,1}, the number of filled pixels is {1,1,1,0,0}, and the output channel dimension is {512,512,256,256,256}. Table 2 displays the complete backbone network parameters.

### 3.2. Multi-Scale Feature Extraction

Literature [29] indicates that the ability of the feature pyramid model to divide-and-conquer at different scales plays the most significant role. In contrast, feature fusion of the model and contextual semantic information fusion is not essential factors in improving detection accuracy. On the basis of this concept, a multi-scale feature extraction structure has been developed to increase the information regarding the multiple scales of detection target features contained in the prediction feature layer. Due to the fixed size of the convolutional receptive field, it can cover different object sizes by stacking dilation convolution with varying sizes of steps. Additionally, a residual connection is used for feature fusion to ensure that the features extracted from the small receptive field are not lost when employing the dilation convolution with a larger receptive field. The completed structure for multi-scale feature extraction is depicted in Figure 4.

The literature [13] proposes a spatial pyramid pooling mechanism that can generate feature representations of a fixed length, making the neural network adaptable to different-sized images. Concurrently, due to the use of multi-level pooling, the model becomes more robust against object deformation, which improves the detection accuracy of the model based on the aggregation of variable scale features. Therefore, when designing the multi-scale feature extraction, we rely initially on the processing of feature pyramids, employing 1 × 1 convolution layers to reduce the channel dimensionality and then 3 × 3 convolution to refine the semantic contextual information. To achieve the same effect of fusing features of different scales as spatial pyramid pooling, the main branch of multi-scale feature extraction is 3-way. The features with different scale receptive fields are generated by stacking two sets of 3 × 3 expanding convolutions with expansion rates of {1,2,3,4} and {2,4,6,8}, respectively. In contrast, the original features are stitched to the feature layer after multi-scale feature extraction using residual connections to achieve the same effect as spatial pyramid pooling. Finally, in the feature extraction using expanded convolution, use 1 × 1 convolution to restore the number of channels. Use identity to make the remaining blocks for feature fusion so that features collected by the shallow network do not get lost when extracting large receptive field features.

### 3.3. Multi-Scale Attention Mechanism

Attention techniques such as SENet [34], CBAM [35], SPANet [36], SKNet [37], and EPSANet [38] have been demonstrated to significantly improve the resolution of difficulties in image processing, natural language processing, etc. The primary characteristic of the attention mechanism is its capacity to favor the most informative feature representation in the assignment to prevent uniform feature representation and to employ global information to emphasize valuable characteristics while suppressing less useful ones selectively. On the one hand, the existing attention mechanism can recalibrate the channel response, spatial response, or both. On the other hand, the attention mechanism is able to adjust the size of the receptive field adaptively. This concept is mainly inspired by the phenomenon of adaptive modulation of receptive field size following visual cortical neuron activation. Furthermore, good model interaction in both the channel and spatial dimensions can often improve model performance while making the model less complicated and enhancing model generalizability.

By extracting multi-scale features, the network model can gather a lot of information about multi-scale features. However, not all of the features increase the model’s detection accuracy. Since X-ray images frequently have complex backgrounds and occlusion stacking between objects, as depicted in Figure 5, the multi-scale feature extraction module contains many redundant background features. A large number of redundant background features and features extracted after the occluded objects will result in severe category imbalance, which is not conducive to improving the accuracy and generalizability. Therefore, it is necessary to design a multi-scale attention mechanism, added after the multi-scale feature extraction module, to improve the information that is beneficial to the accuracy and generalizability. Because both spatial and channel-based attention mechanisms can improve the model’s detection accuracy, the feature layer following the multi-scale feature extraction module contains perceptual receptive field information at each scale. It is necessary to design a multi-scale attention mechanism that interacts with adjacent channels and spaces’ feature information. In addition, when constructing the attention mechanism, we should consider the size of the number of parameters since an excessive number of attention modules can slow down the model’s inference speed when integrated into the framework for target detection. Since the multi-scale attention mechanism is applied to the feature prediction layer, the feature map size in the last two layers of the feature prediction layer, i.e., additional layers 4 and 5, is downsampled to 3 × 3 and 1 × 1, respectively. The feature map no longer has accurate location information, so only the squeeze-excitation (SE) module is used to interact with the channel information in these two layers.

In the implementation, a multi-scale self-attention module is designed, primarily consisting of the spatial self-attention module and the channel self-attention module. The overall local interaction of channel and spatial feature information is attained by processing the input tensor of multiple scales in parallel. Referring to the CBAM design strategy, the channel attention module is similarly positioned ahead of the spatial self-attention module. Figure 6a,b depict the detailed modules. In order to avoid increasing an excessive number of parameters, the multi-scale attention mechanism is constructed using grouped convolution [39] to segregate the feature information at various scales. Compared to the standard convolutional parameter number, the grouped convolution parameter and computation number (Equations (5) and (6)) decrease *g* times. *g* is the number of groups, *k* is the size of the convolution kernel, *C_in_* is the number of input channels, and *C_out_* is the number of output channels. *h_out_* represents the height of the output feature map, whereas *W_out_* represents its width.
(5)$$\begin{array}{l} C_{in}\times k\times k\times C_{out}\\ \frac{C_{in}}{g}\times k\times k\times \frac{C_{out}}{g}\times g \end{array}$$
(6)$$\begin{array}{l} (C_{in}\times 2\times k\times k)\times H_{out}\times W_{out}\times C_{out}\\ (\frac{C_{in}}{g}\times 2\times k\times k)\times H_{out}\times W_{out}\times \frac{C_{out}}{g}\times g \end{array}$$

The size of the previous feature prediction layer *X_i_*_−1_ is adjusted to the following scale of feature prediction layer *X_i_* using a set of 1 × 1 convolution and 3 × 3 convolution. That would make the attention mechanism adapt to the sizes of different feature prediction layers, after which a multi-scale convolution kernel partitions the input feature *X_i_* into parts of *S*, i.e., [*X_i_*_0_*, X_i_*_1_, …, *X_is_*_−1_]. Grouped convolution would handle input features with different convolution kernel scales without increasing the computational effort. The relationship between the number of groups and the size of the convolution kernel scale follows the design Equation (7) of the literature [39], where *G* represents the number of groups and *k* is the convolution kernel size. The final adopted convolution kernel sizes are [3, 5, 7, 9], respectively. The number of groups corresponding to them is [1, 4, 8, 16], and the convolution process is shown in Equation (8), where *k_i_* represents the *i*-th group convolution kernel size. *X_i_* represents the current feature prediction that covers the same receptive field. The process of channel interaction is shown in Equation (9), where *C*_0_ and *C*_1_ represent two 1 × 1 convolution layers, and *σ* represents the sigmoid nonlinear activation function. *H* and *W* represent the feature map’s width and height, respectively. Lastly, the global information after the cross-channel interaction of multiple convolutional kernels is added to the size of the input dimension and then put through a Softmax operation to adjust the attention vector at the channel level and get the multi-scale channel rescaling weights. This is shown in Equation (10), where *G_i_* is the attention weight of each channel.
(7)$$G=2^{\frac{k-1}{2}}$$
(8)$$F_{i}=Conv(k_{i},X_{i}),\;i=0,1,2,\ldots S-1$$
(9)$$G_{i}=\sigma (C_{1}(\sigma (C_{0}(\frac{1}{H\times W}\sum_{u}^{H}\sum_{v}^{W}F_{i}(u,v)))))$$
(10)$$\begin{array}{c} G=Cat([G_{0},G_{1},G_{2},\ldots ,G_{s-1}])\\ att_{i}=\mathrm{Softmax}(G_{i})=\frac{\exp(G_{i})}{\sum_{i=0}^{s-1}\exp(G_{i})} \end{array}$$

After recalibration of the channel-level attention vector, the output tensor generates a spatial attention map utilizing the spatial relationship between features. The spatial attention information is obtained by stitching all channels’ mean and maximum values of the corresponding feature layers. The information regions can be effectively highlighted by applying pooling operations along the channel axes [22] states. Following this, the spatial attention of each scale is fused with the MaxPooling and average pooling information using 7 × 7 window size, with the fill value set to 3 and the stride set to 1 to maintain the size of the feature map. A spatial attention output tensor is generated for each scale of the feature map, as depicted in Equation (12). The spatial attention information is then recalibrated using a sigmoid operation, as illustrated by Equation (13). Experiments show that using a multi-scale attention method can increase the model’s detection *mAP* value by about 2.1% on the dataset.
(11)$$F_{i}=Cat(AvgPool(G_{i}),MaxPool(G_{i})),\;s=0,1,\ldots ,s-1$$
(12)$$F_{t}=Conv_{7}(F_{t},F_{t+1}),\;t=0,1,2,\ldots ,s-2$$
(13)$$F=Sigmoid(F_{t})=\frac{1}{1+e^{-F_{t}}}$$

The final improved target detection framework is shown in Figure 7.

## 4. Experiment

### 4.1. Experimental Environment and Hyperparameter Setting

All experiments with this technique were conducted using an Nvidia RTX3070 (12 G graphics memory) graphics card with a Windows 10 operating system, 32 G.B. of RAM, the CUDA11.6 accelerated computing library, and the Pytorch framework. The batch size was set to 22. To prevent the instability of the model gradient during the initial training, a stochastic gradient optimizer (SGD) with momentum was used for 350 rounds of pre-warm-up training, with the initialized learning rate set to 8 × 10^−6^, the momentum set to 0.9, and the weight decay rate set to 0.0005. After that, using the Adam optimizer with an initialized learning rate of 0.01 to retrain the model. After 2500 epochs, the learning rate automatically lowers to 0.1 times the initial rate. The smoothing constants are, respectively, 0.90 and 0.999, and the weight decay rate is 0.0005. Fifteen thousand epochs make up the entire number of iteration epochs.

Applying data augmentation techniques would improve the robustness and generalization performance of the model for various input image target sizes and shapes. The improved SSD algorithm makes use of data augmentation techniques, such as randomly flipping each training image with a probability of 50% and randomly altering the image color with a brightness of 0.125, contrast of 0.5, saturation of 0.5, and hue of 0.05. Randomly cropping the picture helps improve the detection precision of small-scale objects. The figure depicts the progress of the random cropping algorithm (Figure 8).

### 4.2. Introduction to the Dataset

All the datasets utilized in the tests are from the open source EDS dataset of Beijing University of Aeronautics and Astronautics and MEGVII Research Institute, which comprises 14,219 photos from 3 distinct X-ray machines, 10 classes of objects, and 31,655 target instances in total. Existing X-ray imaging technologies display various colors for detecting various objects. Images of inorganic objects are in shades of blue, images of organic objects are in shades of orange, images of mixtures are in shades of green, and images of things that X-rays can not get through are in shades of red. The detected target’s color, size, and shape are used to label different data sets to tell them apart. The network learns how to perform target detection based on existing labeled data. During the training phase, the data set was randomly picked and separated into a training set of 11,375 images designated EDS_10K and a test set of 2843 images in a ratio of 4:1. Five thousand images were taken from the training set and named EDS_5K to facilitate the ablation experiment. Figure 9 shows the sample distribution of the dataset. In general, the target samples are relatively evenly distributed. The uniformly distributed number of targets enables the target detection network to obtain meaningful feature representation capability without overly favoring certain data, improving model generalization capability. In addition, the image detection targets are classified into three classes based on the COCO dataset classification criteria. Detection targets smaller than 32 pixels are classified as “small targets”, detection targets larger than 32 pixels but less than 96 pixels are classified as “medium targets”, and detection targets larger than 96 pixels are classified as “large targets”. The specific statistical figures are shown in Figure 10. Since targets of different scales exist, it is possible to verify the improved algorithm’s detection accuracy for multi-scale targets.
(14)$$IOU=\frac{area(B_{p}\cap B_{gt})}{area(B_{p}\cap B_{gt})}$$
(15)$$TPR=\frac{TP}{TP+FP}$$
(16)$$FPR=\frac{TP}{TP+FN}$$

### 4.3. Assessment Criteria

To illustrate the model’s detection accuracy, *mAP*, currently the most widely used evaluation index in image processing, is used to measure the detection accuracy of the model. At the same time, the detection speed of the model is measured by the number of parameters as well as the actual inference speed. Intersection over union separates the true negative, false positive, and negative samples. The *mAP* value is calculated by the intersection over union ratio between the prediction result and the true sample. Finally, the prediction boxes are arranged from most confident to least confident. The detection accuracy of the model is shown by calculating the precision rate and the recall rate, and the P-R curve of the model is drawn. The final *mAP* value is obtained by calculating the weighted P-R curve area of each category, as shown in Equations (14)–(16).

### 4.4. Analysis and Comparison of Experimental Results

During the experiment, the loss decrease during the training phase for the EDS_5k dataset is depicted in Figure 10. Compared to the existing mainstream target detection algorithms, the improved SSD algorithm on the EDS_5k dataset achieved 78.7% in *mAP* value. Finally, by training the model on EDS_10k, OPIXray, and DBF6 while keeping the test set unchanged, the *mAP* value improved by 1.9%, and the loss function during training is shown in Figure 11. Adjusting the learning rate to 0.1 times the current learning rate helps hasten the network’s convergence if, during the training of a specific epoch, the network convergence pace begins to slow, and the loss value’s declining trend begins to stall. The model’s final measured number of parameters was 34.82 M, and the inference speed decreased to 48 ms. The improved model significantly increased the effect of detecting small targets, up to 5.1% of the *mAP* value. It improved the effect of detecting medium targets by 8.8% and large targets by 7.5%. Table 3 displays the experimental outcomes of the improved SSD algorithm in comparison to the most prevalent target detection algorithms currently in use. The final results demonstrate that the newly added algorithm outperforms the other fundamental algorithms by a significant margin. Notably, only the information granules in their space are utilized by G-RCNN networks.

The improved model can obtain the highest *mAP* value of 80.6% at an input image size of 300 × 300 while maintaining a high inference speed. The original model can achieve a detection accuracy comparable to the input image size of 512 × 512 (*mAP* value of 80.3%). Still, the inference speed of the original model is only 196 ms due to the sharp increase in the default anchor frames (98112), which is caused by the increase in the input image size. Multiple sets of ablation experiments were conducted on the EDS_ 5K dataset to determine the effectiveness of each module. The network detection accuracy could be improved by 0.8% after replacing the backbone network; the multi-scale feature extraction module could improve the model detection accuracy by 0.9%; the multi-scale attention module could improve the model detection accuracy by 0.7%. Table 4 shows the results of the ablation experiments. The class activation heat map also confirms the multi-scale attention module’s effectiveness by visualizing prediction results. The images are depicted in Figure 12, where the first row of images is the actual image and its label (in the orange box), the second row is the class activation heat map without the multi-scale attention module, and the third row is the improved class activation heat map. The results of the forecast are depicted in Figure 13. In plotting the prediction results, the results after non-maximal suppression for targets with thresholds below 0.4 are filtered out, indicating that the model can recognize all types of objects with high confidence.

X-ray images are frequently densely packed with objects and heavily obscured. Traditional non-maximum suppression methods typically set the confidence of predicted anchor frames with an intersection ratio exceeding a threshold of 0. This frequently results in the loss of overlapping targets. As a result, redundant prediction anchor frames are filtered out using a combination of Adaptive-NMS [40] and Soft-NMS [41] in the post-processing phase. Equation (17) is subsequently used to calculate the target density, where the density of object *i* is defined as the maximum bounding box IoU with other objects in the set of ground truth *G*. Equations (18) and (19) are utilized to execute Soft-NMS, reducing the confidence score of the predicted target frame. *N_M_* represents the adaptive NMS threshold for M, and d_M_ is the object's density that *M* covers.
(17)$$d_{i}:=\underset{b_{i}\in G,i\neq j}{\max}iou(b_{i},b_{j})$$
(18)$$N_{M}:=\max(N_{t},d_{M})$$
(19)$$s_{i}=\begin{array}{c} s_{i},iou(M,b_{i})<N_{M}\\ s_{i}f(iou(M,b_{i})),iou(M,b_{i})<N_{M} \end{array}$$

The feature layer extracted by the backbone network is first reduced using 1 × 1 convolution, followed by splicing the output tensor of the feature prediction layer with the output tensor of the classification network and the anchor frame regression network. A 5 × 5 convolution kernel is used in the last layer of the density network to consider the surrounding information. Finally, the results are mapped to the [0, 1] interval by the Softmax operation. The final output dimension is kept the same as the output dimension of the anchor frame regression network. In particular, since the density prediction network needs to use 5 × 5 convolutional kernels to generate the final results, only the first four feature prediction layers are added to the density prediction network. The density sub-network is shown in Figure 14. The total model detection accuracy was improved by 1.2 mAP values. The specific enhancement results are shown in Table 5.

In addition to further evaluating and validating the effectiveness of the proposed multi-scale attention module and multi-scale attention module, we conducted additional experiments on the DBF6 and OPIXray datasets. The DBF6 dataset contains six types of tools with varying degrees of occlusion, and the OPIXray dataset contains five types of stacked tools. In the DBF6 dataset, FA denotes firearm, FP denotes firearm part, CA denotes camera, KN denotes knife, CK denotes ceramic knife, and LA denotes laptop. In the OPIXray dataset, FO represents Folding Knife, ST represents Straight Knife, SC represents Scissor, UT represents Utility Knife, and MU represents multi-tool Knife. In a ratio of 4:1, each dataset was divided into training and test sets. On the OPIXRay dataset, the training set contains 80% of the images (7109), and the test set contains 20% (1776). On the DBF6 dataset, the training set contains 12,360 images, and the test set contains 3089 images. During the experiment, we uniformly scale the images to 300 × 300 size before feeding them into the neural network. All models were optimized by the SGD optimizer with the learning rate set to 0.0001. The batch size was set to 22, and the momentum and weight decay were set to 0.9 and 0.0005, respectively. We evaluated the average mean precision (mAP) of the target detection to measure the performance of the models, and the IOU threshold was set to 0.5.

In the OPIXRay dataset, the literature [22] proposes a de-obscuring attention module (DOAM) to capture edge information and area information separately. SSD, YOLOv3, and FCOS network architectures were selected as baselines into which the DOAM modules were incorporated to verify the improvements. The same baseline was chosen to fuse the MSA and MSE coupling modules separately when performing comparison tests. In the DBF dataset, the original paper focused on transfer learning of X-ray images and used Faster-RCNN and R-FCN networks as baselines. In the same way, we used the two network architectures above as a baseline to compare the work of our proposed modules. The experimental comparison results demonstrate that the proposed modules can significantly improve the detection precision of all categories on the DBF6 dataset. The detection accuracy of all categories exceeded 90% except for Knife, a tiny target. The improved SSD model on the OPIXray dataset outperformed the DOMA architecture reported in the literature [22] and attained the best detection accuracy of 82.9% mAP. The performance of multi-scale feature extraction and multi-scale attention integration network on YOLOv3 and FCOS improved the detection accuracy by 0.9% and 0.2%, respectively. The experimental results of DBF6 dataset are shown in Table 6, and OPIXray dataset are shown in Table 7. The detection results are shown in Figure 15.

## 5. Conclusions

In this research, we propose an enhanced multi-scale target detection method based on the SSD algorithm, which seeks to increase the detection accuracy of the model for each scale target while ensuring the model’s detection speed. We use the EDS dataset as our primary training dataset. With approximately 22 pictures per second, the current mainstream model achieves its highest detection accuracy. The modified Resnet 50 network architecture replaces the original VGG network to improve the feature extraction capability of the backbone network while reducing the number of network parameters. A multi-scale feature extraction module is designed with stacked convolutional kernels of different sizes. The algorithm’s performance is further enhanced by maximizing the feature expression potential of the output feature map of the backbone network. In addition, a multi-scale attention mechanism is intended to improve the detection impact by concentrating feature information at various scales in both channel and spatial dimensions. Ablation experiments demonstrate that adding multi-scale feature extraction and multi-scale attention coupling mechanisms into a model improves its detection accuracy compared to employing only one of the modules. Ultimately, we combine our newly developed multi-scale multi-feature extraction and attention coupling modules with other target detection models on different X-ray datasets. The results of the experiments show that the new architecture works well with other datasets.

## 6. Discussion

This paper provides a reliable solution for intelligent security screening. This paper proposes a method with better results than the fast detection speed but low detection accuracy of the YOLO series and the slow detection speed of the original SSD algorithm. The idea is mainly to consider that authentic images will generally contain targets of various scales in real life. It is often difficult to adapt to targets of different scales by using a fixed-size convolution kernel. It can be seen that this study uses ResNet_50 as the backbone network to save the computational overhead in network inference as much as possible. Reasonable speculation is that if a deeper neural network such as ResNet-101 or Res2Net-101 is used as the backbone network, it can further improve the detection accuracy of the model. In addition, the model can still be further improved in terms of the number of parameters and detection speed, and a more lightweight model still needs to be researched in the future. In this paper, the contrast of X-ray images is made higher during data enhancement so that the existing network can directly predict targets in different color gamuts. However, the network architecture for detecting targets in different color gamuts separately still needs more research.

## Figures and Tables

**Figure 1 sensors-22-07836-f001:**
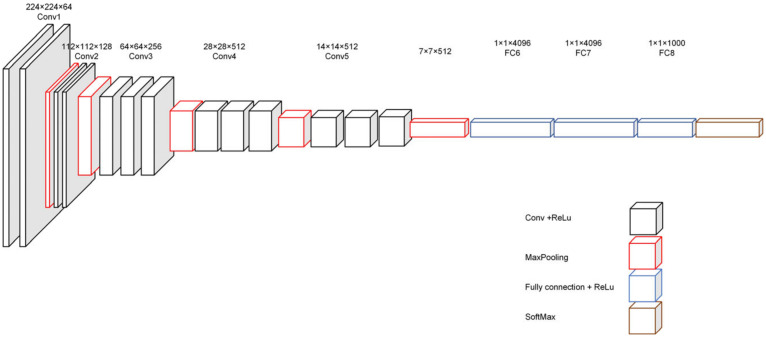
VGG—16 network architecture diagram.

**Figure 2 sensors-22-07836-f002:**
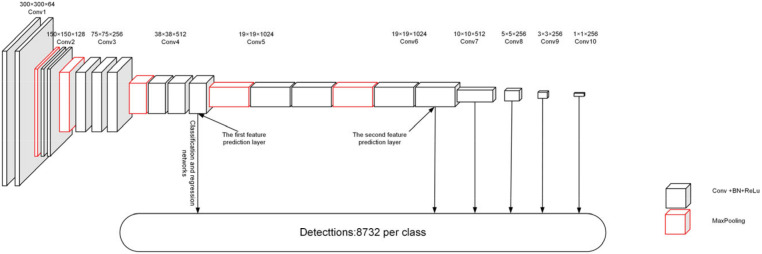
SSD approach backbone network.

**Figure 3 sensors-22-07836-f003:**
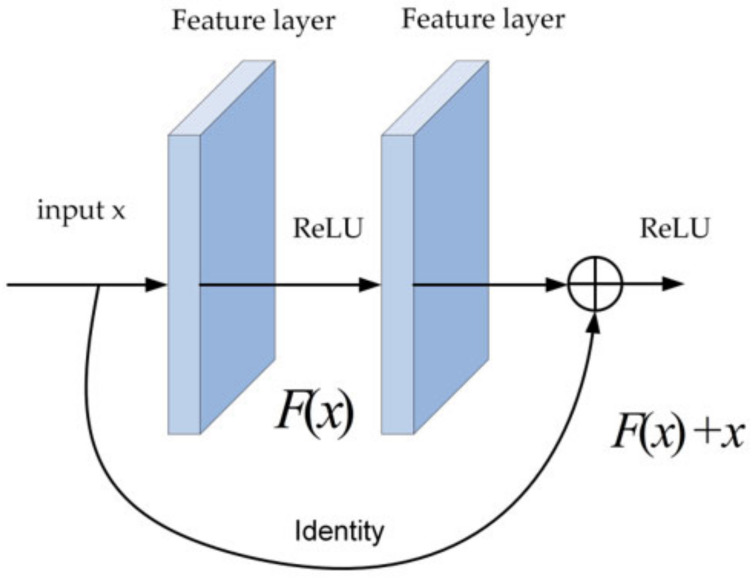
Residual block.

**Figure 4 sensors-22-07836-f004:**
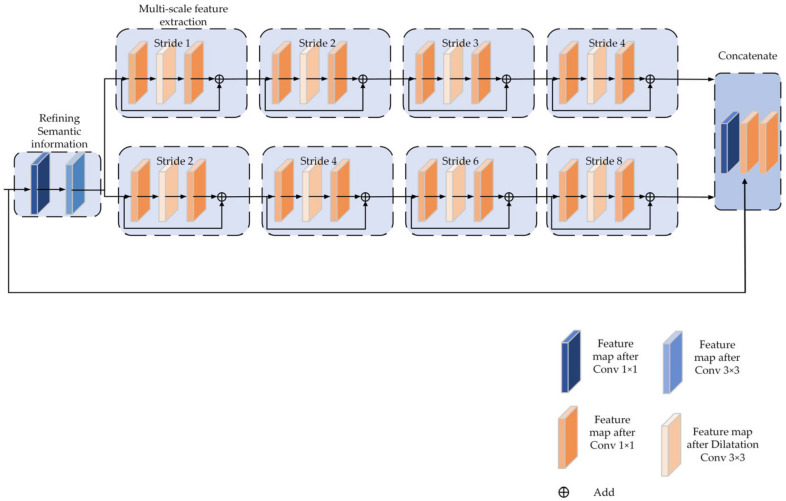
Multi-scale feature extraction module.

**Figure 5 sensors-22-07836-f005:**
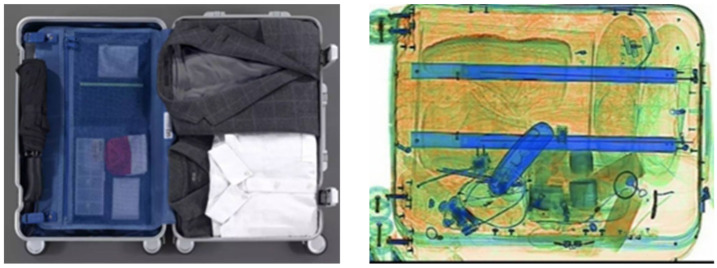
(**Left**) Normal scene. (**Right**) X-ray scene.

**Figure 6 sensors-22-07836-f006:**
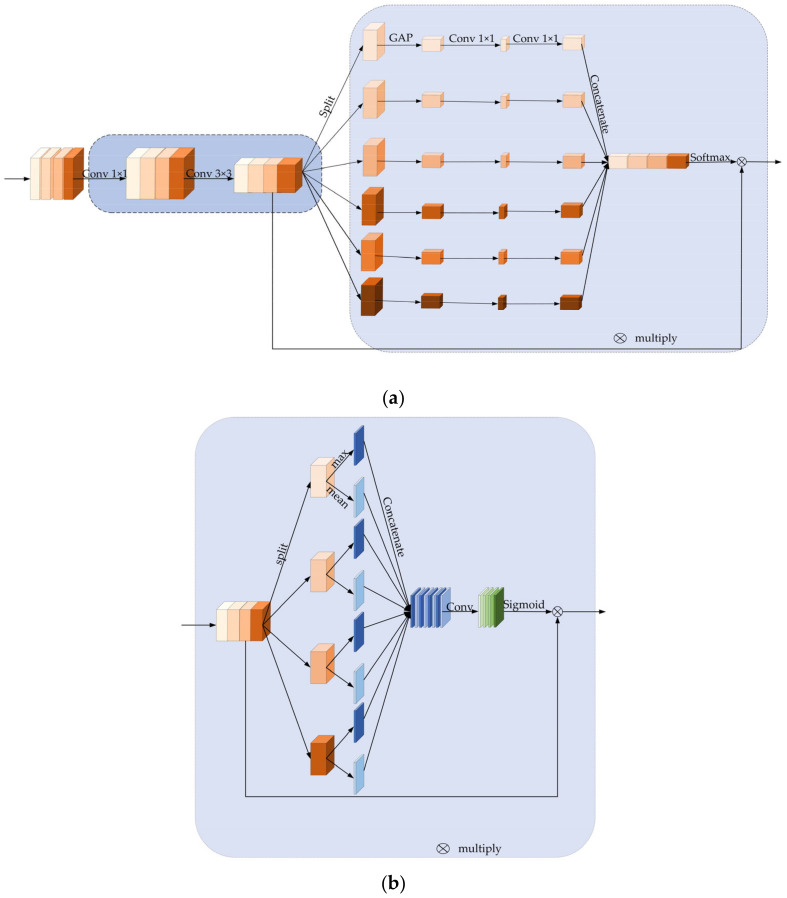
(**a**) Multi-scale attention mechanism (channel). (**b**) Multi-scale attention mechanism (space).

**Figure 7 sensors-22-07836-f007:**
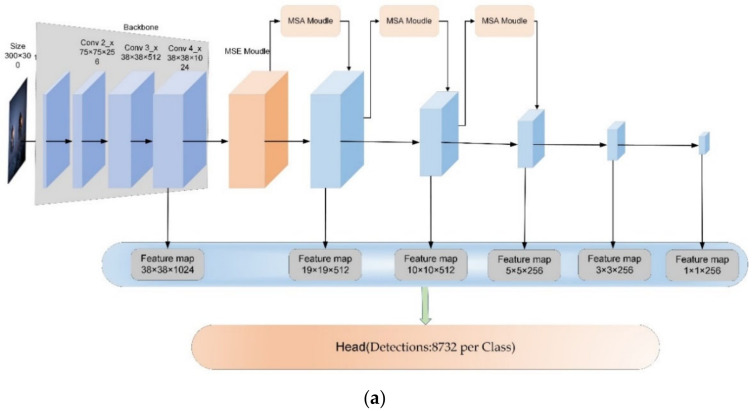
(**a**) Improved target detection framework. (**b**) Objective classification network with regression network (prediction head).

**Figure 8 sensors-22-07836-f008:**
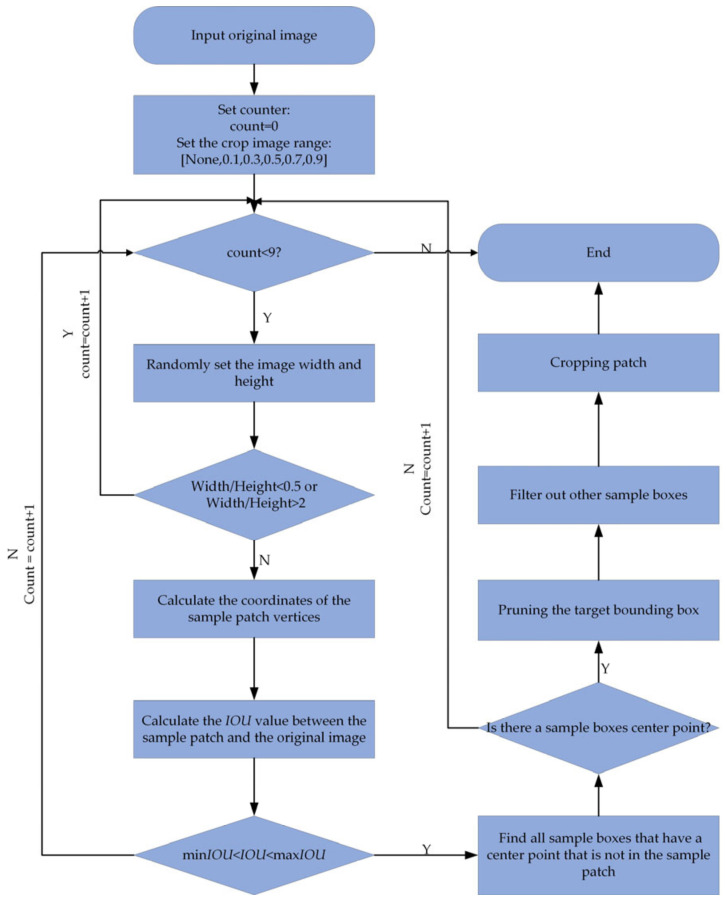
Flow chart of random cropping algorithm.

**Figure 9 sensors-22-07836-f009:**
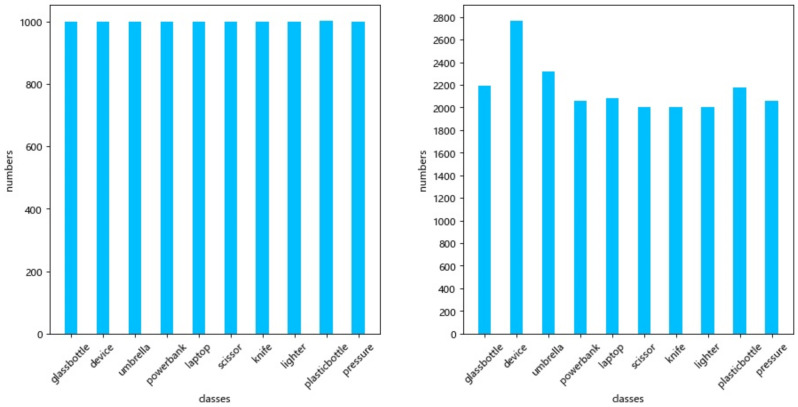
(**Left**) Number of target instances in EDS_5k. (**Right**) Number of target instances in EDS_10k.

**Figure 10 sensors-22-07836-f010:**
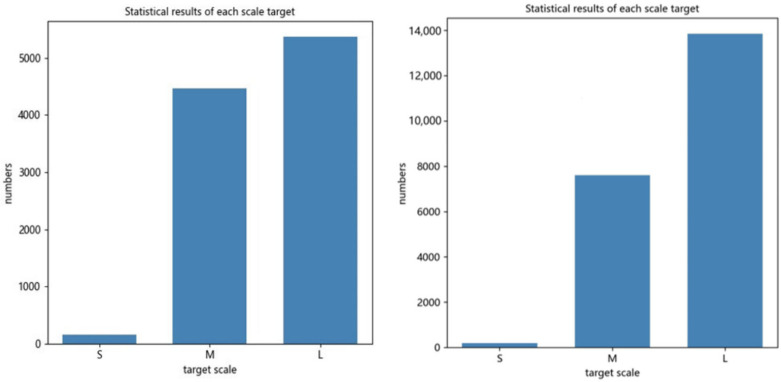
(**Left**) Distribution of targets by scale for EDS_5k. (**Right**) Distribution of targets by scale for EDS_10k.

**Figure 11 sensors-22-07836-f011:**
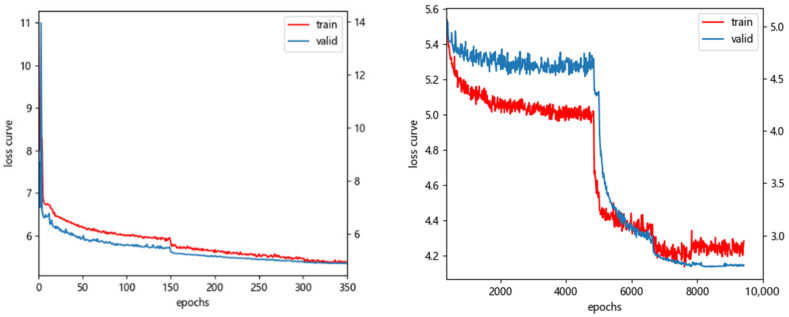
(**Left**) Warm-up training. (**Right**) Official training.

**Figure 12 sensors-22-07836-f012:**
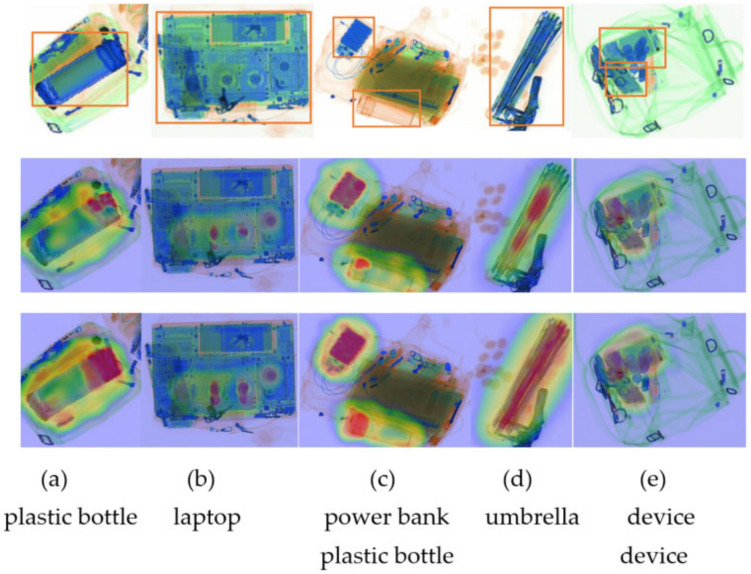
Visualization results of the class activation map.

**Figure 13 sensors-22-07836-f013:**
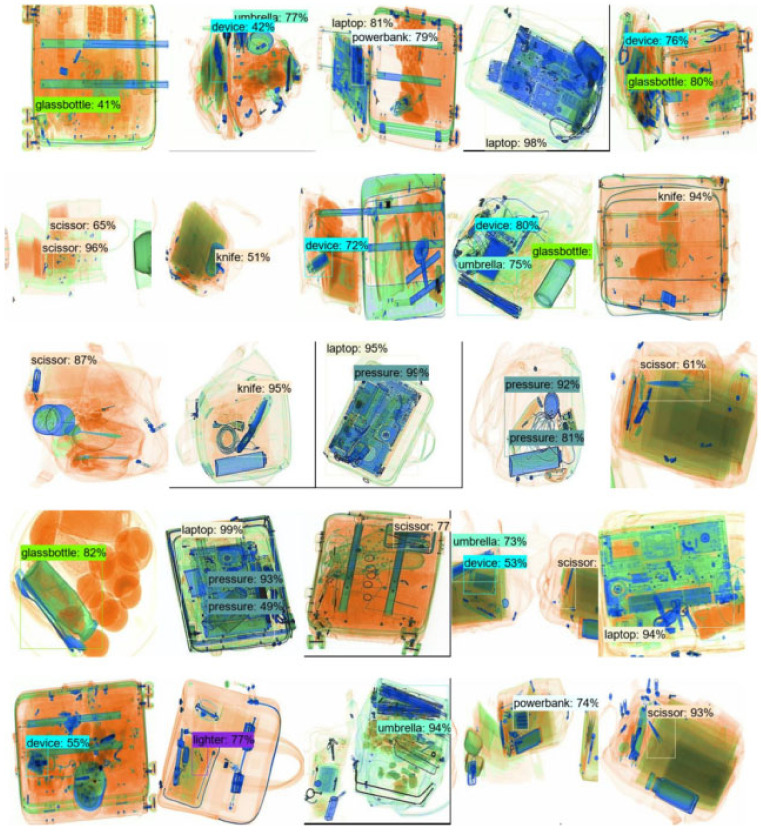
Prediction effect.

**Figure 14 sensors-22-07836-f014:**
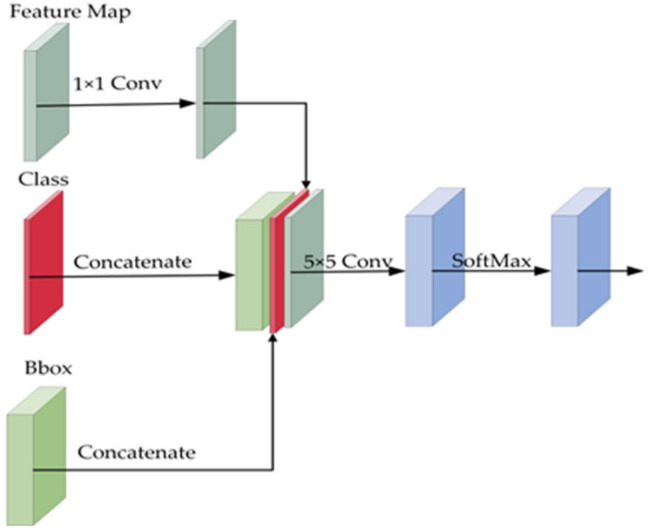
Density subnet.

**Figure 15 sensors-22-07836-f015:**
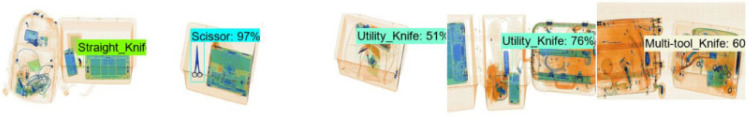
OPIXray data set prediction results.

**Table 1 sensors-22-07836-t001:** Predefined anchors and aspect ratios.

Feature Map Size	Anchor Boxes Size	Aspect Ratios
38 × 38	21, 45	1, 2, 0.5
19 × 19	45, 99	1, 2, 0.5, 3, 1/3
10 × 10	99, 153	1, 2, 0.5, 3, 1/3
5 × 5	153, 207	1, 2, 0.5, 3, 1/3
3 × 3	207, 261	1, 2, 0.5, 3, 1/3
1 × 1	261, 315	1, 2, 0.5, 3, 1/3

**Table 2 sensors-22-07836-t002:** Backbone network parameters.

Layer Name	Output Size	VGG	ResNet
Conv1	150 × 150	[Conv3] × 2	[Conv7] × 1
Conv2	75 × 75	[Conv3] × 2	[Conv1, Conv3, Conv1] × 3
Conv3	38 × 38	[Conv3] × 2 [Conv1] × 1	[Conv1, Conv3, Conv1] × 4
Conv4	38 × 38	[Conv3] × 2	[Conv1, Conv3, Conv1] × 6
Additional layer1	19 × 19	[Conv1, Conv3] × 1	[Conv1, Conv3] × 1
Additional layer2	10 × 10	[Conv1, Conv3] × 1	[Conv1, Conv3] × 1
Additional layer3	5 × 5	[Conv1, Conv3] × 1	[Conv1, Conv3] × 1
Additional layer4	3 × 3	[Conv1, Conv3] × 1	[Conv1, Conv3] × 1
Additional layer5	1 × 1	[Conv1, Conv3] × 1	[Conv1, Conv3] × 1

**Table 3 sensors-22-07836-t003:** Comparison of mainstream methods.

Approach	*mAP* _0.5_	*mAP* _0.75_	*mAP_s_*	*mAP_m_*	*mAP_l_*	Inference Speed (ms)
Fast-RCNN	67.2	45.7	57.8	71.5	72.3	396
Faster-RCNN	69.9	46.3	58.3	74.9	76.5	142
G-RCNN	77.3	55.1	**61.4**	83.9	88.4	89
YOLOv4	75.1	53.5	49.7	88.4	87.2	33
YOLOv5	75.5	55.5	49.7	87.8	89.3	**28**
YOLOv7	77.2	55.5	52.3	88.7	90.6	37
SSD300	74.4	51.9	47.5	85.3	87.4	52
DSSD321	75.9	53.3	55.1	85.1	87.5	67
Improved SSD300	**78.7**	**56.0**	52.6	90.3	93.2	47
Improved SSD300 (10k)	**80.6**	**56.3**	52.6	**94.1**	**95.1**	47

**Table 4 sensors-22-07836-t004:** Ablation experiments.

Approach	Backbone	*mAP*
SSD300	VGG16	74.4
SSD300	ResNet50	75.2
SSD + MSE	ResNet50	76.1
SSD + MSA	ResNet50	77.5
SSD + MSE + MSA	ResNet50	78.7

**Table 5 sensors-22-07836-t005:** The prediction results of different non-maximum suppression.

NMS Approach	*mAP* _0.5_	*mAP* _0.75_	*mAP_s_*	*mAP_m_*	*mAP_l_*
greedy NMS	80.6	56.3	52.6	94.1	95.2
Adaptive-NMS	81.1	55.9	52.6	95.1	95.6
Adaptive-NMS + Soft-NMS	81.8	58.0	52.9	95.8	96.7

**Table 6 sensors-22-07836-t006:** DBF6 contrast experiments.

Baseline	Method	Dataset	*mAP* _0.5_	FA	FP	CA	KN	CK	LA	Inference (ms)
Fast-RCNN	Transfer Learned	DBF6	85.1	91.6	90.1	84.4	67.7	88.9	87.9	150
Fast-RCNN	MSA + MSE	DBF6	88.3	92.1	92.5	90.1	71.9	91.4	91.8	175
R-FCN	Transfer Learned	DBF6	85.6	94.2	92.5	88.7	55.6	92.0	90.6	85
R-FCN	MSA + MSE	DBF6	89.5	94.7	92.9	91.7	65.9	96.6	95.2	93

**Table 7 sensors-22-07836-t007:** OPIXray contrast experiments.

Baseline	Method	Dataset	*mAP* _0.5_	FO	ST	SC	UT	MU	Inference (ms)
YOLOv3	DOAM	OPIXray	79.3	90.2	41.7	97.0	72.1	95.5	23
YOLOv3	MSA + MSE	OPIXray	80.2	91.3	41.7	97.8	74.3	95.9	30
FCOS	DOAM	OPIXray	82.4	86.7	68.6	90.2	78.8	87.7	26
FCOS	MSA + MSE	OPIXray	82.6	86.9	69.3	90.2	80.3	85.7	34
SSD	DOAM	OPIXray	74.1	81.4	41.5	95.1	68.2	84.3	55
SSD	MSA + MSE	OPIXray	82.9	92.1	59.7	97.5	73.3	91.9	41

## Data Availability

Not applicable.

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
