# Peer review of "Multi-Object Detection in Security Screening Scene Based on Convolutional Neural Network"

_sensors, 2022, doi:10.3390/s22207836_

Round 1

Reviewer 1 Report

This paper presented a coupled structure of multi-scale feature extraction, and fused multi-scale attention mechanism is mainly fused on the basis of one shot multi-box detection architecture. The contrast and ablation experiments showed that the proposed algorithm has better performance. However, there is a large number of grammatical errors in this paper, which need to be improved. Rejection is needed. Related comments and suggestions are below:

1.      The article’s structure is unreasonable, and there is no discussion section. Besides, this paper mainly studies the target detection of X-ray images, the development of one-stage target detector and multi-stage target detector models took up more space in the Introduction, which is not the main focus of your research. It is suggested to complement the research development on target detection of X-ray images and the research gaps in the previous algorithms.

2.      Some professional terms are misused in this paper, such as One Shot MultiBox Detector (SSD) and sensory field.

3.      For deep learning or machine learning, the number of training samples, and test samples are important. In Section 3.2, how do you name the training and test sets? What’s the meaning of 5k/10k/1k, respectively? Typically, the training set and test set are divided according to the ratio of 7:3 or 4:1. However, no description of that is found in your paper. It is suggested to provide a more detailed explanation.

4.      A multi-scale feature extraction is added to the detection module to adapt to targets of different scales. However, in the experimental part, no results can intuitively see the detection ability of the model for targets of different scales. It is suggested that the author increase relevant experiments to verify the detection results of the model for small, medium, and large targets.

5.      X-ray images have the limitation of low color contrast, the input image must be given false colors to distinguish objects, which means that there may be color differences between two images of the same object, or different objects are of the same color. The author can discuss how to deal with such problems in this paper and briefly discuss the impact of image colors on recognition accuracy.

6.      The conclusions should be presented in a more general form, emphasizing what new data was obtained, how it broadened our knowledge, and whether it made a new contribution to the methodology of studying detection. It is also worth writing about future research directions.

Reviewer 2 Report

In this paper, the authors have developed a deep learning framework to improve the detection accuracy of X-ray images in security screening scenarios. They used Resnet-50 as a backbone network, and a multi-scale feature extraction (MSE) structure was designed to enhance the information in the multi-stage prediction feature layer. To prevent redundant feature interference and extract effective contextual information, the multi-scale attention mechanisms (MSA) architecture is integrated on the prediction feature layer. In the experiment, they have used the open-source EDS dataset and the results show that the proposed method gives a promising outcomes. The work the author has presented is novel enough, however there are some concerns that need to be addressed.

Comment 1: The author has used Resnet-50 as a backbone network, but other improved ResNet versions, in particular Resnet-101, are already available and can significantly boost performance when used as a backbone network. Why is Resnet-50 being used as a backbone network?

Comment 2: The authors are suggested to compare the proposed method with the following papers which use Resnet-101 as a backbone:

1.     K. He, G. Gkioxari, P. Dollár, and R. Girshick, ‘‘Mask R-CNN,’’ in Proc. IEEE ICCV, Dec. 2017, pp. 2980–2988.

2.     N. Bodla, B. Singh, R. Chellappa, and L. S. Davis, ‘‘Soft-NMS— Improving object detection with one line of code,’’ in Proc. IEEE Int. Conf. Comput. Vis., Oct. 2017, pp. 5561–5569.

Also, compare with this recent method:

3.     A. Pramanik, S. K. Pal, J. Maiti and P. Mitra, "Granulated RCNN and Multi-Class Deep SORT for Multi-Object Detection and Tracking," in IEEE Transactions on Emerging Topics in Computational Intelligence, vol. 6, no. 1, pp. 171-181, Feb. 2022, doi: 10.1109/TETCI.2020.3041019.

Comment 3: Only the object with the highest classification score is chosen by non-maximum suppression; otherwise, the object is ignored. This causes less accuracy. There are some improved versions of non-maximum suppression method presented in literature [4-5] which could be adopted for improving the results. 

4.     L. Tychsen-Smith and L. Petersson, ‘‘Improving object localization with fitness NMS and bounded iou loss,’’ in Proc. IEEE Conf. Comput. Vis. Pattern Recognit., Jun. 2018, pp. 6877–6885.

5.     S. Liu, D. Huang, and Y. Wang, ‘‘Adaptive NMS: Refining pedestrian detection in a crowd,’’ 2019, arXiv:1904.03629

Comment 4: The visual quality of figures should be improved. The text written in figures 5 (a)-(b), and 6 (b) are not clearly visible. The equations given in the paper are not well formatted.

Reviewer 3 Report

The manuscript deals with X-ray security imaging. The authors focused on the SSD approach for object detection. The proposed deep network combines a backbone based on ResNet, a multi-scale feature extraction, and multi-scale attention mechanism.

As a general comment, the proposed deep network seems to exploit well-known modules. Thus, the paper may merit publication only if the authors are able to show that the network can reach satisfactory results on the application at-hand. In this regard, the authors should use as reference the following recent survey

https://doi.org/10.1016/j.patcog.2021.108245

The survey identifies X-ray security imaging datasets that are widely used in the literature. The authors should include those benchmarks in the experimental session. Moreover, the authors should thoroughly compare the proposed solution with the state-of-the-art approaches dealing with X-ray security imaging. The comparison should involve classification performances and computational performances

Further comments:

- Sec. 2 now mixes background (e.g., VGG) and the design of the proposed network. The authors should split these two parts. I would suggest adopting this structure: Sec. 3 for the design of the proposed network and Sec. 4 for the experimental result.

- The level of the English language can be improved. In particular, I would suggest using shorter sentences.    

Reviewer 4 Report

This paper proposes a CNN-based network together with multi-scale feature extraction and attention mechanism for detecting multiple objects in x-ray security screening. The network architecture is based on the SSD framework. The backbone network is replaced with ResNet (originally VGG) to obtain more depth. In theory, deeper networks can extract richer semantic information. Multi-scale feature extractors such as FPN can enhance the detection of objects of varying sizes, especially relatively smaller ones. The attention mechanism works by emphasizing some parts of the input data while diminishing the other parts.

1. The authors have written a thorough methodology and provided justification for all their design choices. However, there seems to be some missing context and contradicting statements.

For instance, the authors often refer to the "real-time requirements" of the application as one of the main drivers for choosing a single-shot framework over anchor-based frameworks. Can the authors provide a more specific number to this claim? What is the average detection time required in x-ray security screening? The significance of this work would be more appreciated if the target value is more specific than relative to other algorithms.

2. In Section 2.2.1, the authors describe the different features extracted from different depths in a neural network (e.g., edge contours -> color -> texture -> semantic). The statement implies that different layers of the neural network extract different features. However, in Section 2.2.3, the authors claimed that using multi-scale feature extraction introduces redundant features, so an attention mechanism is needed to suppress these redundant features. If the features from different layers are different, how can there be redundancy? Visualization of where the attention of the network before and after the addition of the attention mechanisms would be persuasive proof. Usually, papers that use attention mechanisms visualize their results using class activation methods.

3. In Section 3.4, the authors mention the sudden drop in the loss during training on one of the datasets but did not provide a follow-up explanation. What caused this drop?

4. In their conclusion, the authors claim that the main idea is to consider real life, wherein it is often difficult to adapt to targets of different sizes. However, the paper does not show results confirming that their method achieved this goal. Although the performance generally improved, it is not clear which object sizes had the greatest benefit. In fact, the authors have already presented a distribution of the objects based on the sizes predefined by the COCO dataset. It would be more compelling if they could also present the mean average precision (mAP) for the small, medium, and large targets, respectively.

5. Besides, this paper also has some typography and formatting issues.

·      Please be mindful of properly spelling author names in citations (see Introduction citation number 24)

·   In Figure 3, what is RULE? Shouldn't it be ReLU, which stands for Rectified Linear Unit?

·         In Figure 4, why are there separate block representations for the same operation (Conv 1x1)?

·         The labels in some of the figures are hard to read, even when zooming in.

Round 2

Reviewer 1 Report

Although there is not much novelty in the paper, I think it can be published.

Reviewer 2 Report

1.      The authors have provided a reasonable rebuttal for all the given comments.

2.      The authors are asked to compare with the following recent method; however, they have not addressed the reason for not comparing this method with the proposed method.

A. Pramanik, S. K. Pal, J. Maiti and P. Mitra, "Granulated RCNN and Multi-Class Deep SORT for Multi-Object Detection and Tracking," in IEEE Transactions on Emerging Topics in Computational Intelligence, vol. 6, no. 1, pp. 171-181, Feb. 2022, doi: 10.1109/TETCI.2020.3041019.

3.       The authors are asked to explore the improved versions of non-maximum suppression methods as a post processing step to increase the detection accuracy. However, they have stated that they have solely focused on evaluating the efficacy of a multi-scale feature extraction and multi-scale attention coupling mechanism.

Reviewer 3 Report

As I already wrote in my first review, the proposed approach does not seem to introduce novel elements. The comparison with the state-of-the-art is indeed not sufficient. The authors added two benchmarks in this new version of the paper. However, they did non explain the setup of the new experiments. Are they comparing the proposed approach with the best state-of-the-art approaches? Are they adopting a standard experimental setup when making such comparison (e.g., size and composition of the training set and test set)?  

In my opinion, the paper is not ready for publication on a journal. The contribution is incremental and there is no evidence that the proposed pipeline can really outperform state-of-the art approaches. 

Reviewer 4 Report

In the revised paper, the authors have included the requested class activation heat map visualization and experiment results for objects of different scales. These supplements have strengthened the claim for using attention mechanisms and multi-scale feature extraction, respectively. However, the author's response to the first comment is still unsatisfactory. Real-time requirements can be different for different applications. For instance, autonomous driving can benefit from hundreds of FPS detection. It seems that the cited minimum of 10 FPS refers to a more general detection case. This paper deals with a specific application. Hence, the baselines must also be presented specifically. What is the average detection time required in x-ray security screening?
